# Changes in Dog Behaviour Associated with the COVID-19 Lockdown, Pre-Existing Separation-Related Problems and Alterations in Owner Behaviour

**DOI:** 10.3390/vetsci10030195

**Published:** 2023-03-04

**Authors:** Emila-Grace Sherwell, Eirini Panteli, Tracy Krulik, Alexandra Dilley, Holly Root-Gutteridge, Daniel S. Mills

**Affiliations:** 1School of Life Sciences, University of Lincoln, Lincoln LN6 7DL, UK; 2Department of Behavior & Training, Humane Rescue Alliance, Washington, DC 20011, USA

**Keywords:** attachment, attention-seeking, behaviour, COVID-19, dog, lockdown, owner, routine, separation anxiety, welfare, work

## Abstract

**Simple Summary:**

During the COVID-19 pandemic, many countries went into lockdowns, this raised concerns about dog behaviour, especially in relation to separation-related problem behaviour. We undertook a monthly survey during which we asked people about their work patterns, dog management practices and their dogs’ behaviour. We found that dogs who showed signs potentially indicating separation-related problems prior to COVID seemed to be more likely to worsen and develop further behavioural issues during lockdown. These changes were not just associated with separation-related issues but included more general issues related to stress. Dogs who, when separated from their owner pre-COVID, vocalized, self-injured, or showed frustration at confinement were particularly prone to developing issues related to owner attempts to go or be out of sight at home during lockdown, e.g., when the owner tried to leave the room. Changes in management appeared to be associated with specific forms of stress with related behaviour changes, for example, changes to a dog’s safe space seemed to result in efforts by the dog to increase control over its environment. Different patterns relating to the emergence over time of the risk of aggression towards the owner were apparent between those working from home and those continuing to work outside the home.

**Abstract:**

During the COVID-19 pandemic, lockdowns provided an opportunity to assess what factors, including changes in an owner’s routine and time spent at home, were associated with changes in dog behaviour. We undertook a longitudinal survey over a period of 8 months during which we asked about people’s work patterns, dog management, and their dogs’ behaviour. Generalized linear models revealed that the pre-existence of signs of potential separation-related problems, and especially vocalisation, self-injury, and chewing to escape confinement, was associated with an increase in a range of separation issues. Dogs showing separation-related signs prior to COVID were also more likely to develop more problems during lockdown. Management changes tended to result in increased physical and social stress, with a range of potential compensatory actions taken by the dog, however these signs of stress did not generally appear to be connected to separation-related issues. Survival analysis was used to investigate the emergence of specific issues over time. This indicated that a change to working from home was related initially to a decreased risk of aggression towards the owner, but over time, those who continued to work from the home were at an increased risk of this problem. No other significant time-related relationships were found.

## 1. Introduction

Following the outbreak of the COVID-19 pandemic (SARS-CoV-2 virus) in December 2019, many countries imposed restrictions on their citizens in order to reduce infection rates [1]. Lockdowns were globally enforced with varying restrictions. Many people changed to working from home, interaction with others was often banned, and schools closed, with many countries only permitting people to leave their homes to carry out essential work. For example, the UK government implemented national lockdown restrictions on 23 March 2020. People were instructed to stay at home and only leave the house for essentials, such as food or medication, and undertake one short period of local exercise a day [2]. By contrast, in the US, lockdown restrictions were more phased. There, the first lockdowns began 15 March 2020, when New York City closed public schools, then on March 19th, California became the first state to issue stay-at-home orders. The federal government never issued lockdown orders; however, 46 states and the District of Columbia temporarily closed all non-essential businesses by the beginning of April of 2020, as per White House Coronavirus Task Force recommendations. In May of 2020, the CDC introduced phased re-opening guidelines. However, schools in 48 states remained closed until September of 2020, when many went to hybrid learning models of both distance learning and in-person classes. Restrictions changed many lifestyles and routines [3,4], including those of dog owners and their dogs [5,6,7]. Since people were encouraged to stay at home and avoid contact with others [8], both the frequency and duration of time that many dogs were left alone at home decreased [4]. Indeed, Christley and colleagues [4] reported that during the UK lockdown there was a 15% increase in the number of dogs that were left alone for less than 5 min and a 42.6% decrease in the number of dogs left alone for 3 or more hours. However, in some cases, such as with key workers, routines remained largely unchanged or working hours lengthened, increasing the amount of time that dogs were left alone [9]. How these changes affected dogs remains unclear.

During lockdown, many pets received more companionship and attention [10], many dogs had more opportunities for play, training, enrichment, and other mental stimulation [4,5]. While this might be beneficial, these changes in routine can also be stressful [9] and they may result in a reduction in quiet time and opportunities for rest by dogs [4]. Previous studies also suggest that there was great variability in lockdown activity levels for dogs, with some owners increasing and others decreasing the frequency, duration, and/or use of leads on walks [4]. For many, the location of walks changed from rural to urban locations, closer to people’s homes [7]. Several previous studies [4,5,11,12,13] have indicated that many owners believed their dogs’ quality of life to have been reduced by the pandemic and were worried by the emergence of novel and often undesirable behaviours. To this end, it is clear that lockdown was associated with several detrimental changes in the behaviour of many dogs, which might impact the human–dog relationship. For example, although Bowen et al. [11] reported increased interaction and a stronger emotional bond between owners and their dogs, a small proportion (5.8%) reported a deterioration in their relationship during lockdown (by comparison 28.8% reported an improvement). The most common behaviour problems reported to worsen were vocalisation (about a quarter of subjects) followed by over 15% increasing reactivity to loud or unexpected noises and over 10% reporting a worsening of social interactions with other dogs when out. Less than 5% reported a deterioration of social behaviour (increased aggressivity) towards family members. By contrast, one qualitative study [14] has reported a wide range of changes in both dog–dog (both aggressivity and desire to interact positively) and dog–human (anxiety and reactivity as well as attention-seeking behaviour and separation-related issues) interaction as well as non-social behaviour: regression of trained responses focused on reducing reactivity to moving objects. Other studies [4,10,15,16] highlight intensifications in social interactions as a result of owners being home, with approximately a third of dogs seeming to follow their owners more during lockdown, and about a quarter appearing more affectionate [15]. Thus, increased contact time between owners and dogs is not universally positive, with increased time together potentially providing a fertile ground for the inadvertent conditioning of social behaviours and subsequent problems. The potential importance of this is further supported by reports of greater owner dependence [6,15,17,18] during lockdown.

In particular, there has been concern over the development of separation-related problems, given the marked reduction in time dogs were spending alone during lockdown [17]. Indeed, the COVID lockdown has also provided a unique opportunity to observe how problem behaviours might develop in response to changes in management. One key study by Harvey and colleagues [17] reported that around 10% of dogs appeared to develop separation-related problems through lockdown; by contrast, nearly half of dogs showing signs of these problems prior to COVID ceased to display them during lockdown. The presence of separation-related problems pre-lockdown was the best predictor of their occurrence in October of that year when restrictions were lifted; but reduced number of days left alone during lockdown was also an important predictor. Older dogs, but not puppies, were also at the greatest risk of developing separation-related problems. There was also a suggestion that the size of the reduction in time left alone was an important predictor of the resolution of signs among those dogs with separation-related problems at the start of lockdown; with those dogs experiencing the smallest change having the greatest resolution of problems. Disruption to routine might therefore have been an important stressor for dogs at this time. These authors also highlight the value of further longitudinal work, which is notably absent from the current literature.

Therefore, our first aim was to investigate the effect of pre-existing conditions (including both pre-existing management practices and problem behaviours in the dog), as well as changes to some pre-existing management-related variables alongside other demographics on the subsequent behaviour of dogs during lockdown.

Our second aim was to use our unique longitudinal dataset to compare the emergence of potentially problematic behaviour in dogs over time. This comparison was based on both whether owners were working from home and the time the dog was left alone during the pandemic, given the increase in working from home at this time and the associated disruptions to normal routines.

## 2. Materials and Methods

All statistics were performed in SPSS 28 (IBM Corp. 2022, Armonk, NY, USA).

### 2.1. Survey Content and Schedule

The data were gathered using an online questionnaire using the online platform, Qualtrics. This questionnaire was divided into three parts:

Part 1 gathered information about the owner and their dogs (with details potentially provided for up to 6 dogs), the presence of other pets in the home, and whether any members of the family had switched to working from home and, if so, on what date. As the survey was international and thus included a range of different starts for the lockdown, the period the dogs had been in lockdown was calculated for each of the participants using their answer to the date that they started to work from home, or if not working from home, the date of the first survey.

Part 2 of the questionnaire related to one dog’s behaviour when left without human company prior to lockdown. As many owners had multiple dogs which were treated roughly the same, the owners were asked to choose to answer for one dog on the basis of the name that came first in the alphabet. The occurrence of eight behaviours was scored by respondents in response to the question: “When left without human company prior to the lockdown, how often did your dog show the following behaviours if left for more than 1 h” using a six-point ordinal rating scale relating to frequency in the given context (every time, most of the time, about half the time, less than half the time but quite frequently, rarely, never). Table 1 lists the behavioural options and their occurrence. The choice of these behaviours was based largely on behaviours related to separation-related problems described by Blackwell and colleagues [19] and de Assis and colleagues [20].

A further set of questions asked about how the dogs were managed prior to COVID-19 lockdown including time left alone, exercise patterns, leaving routines, number of walks, etc., which were also rated in this section. Some of these factors were used to predict the occurrence of problem behaviour at the time of the survey (fixed metrics Table 2).

Part 3 of the survey focused on how things were currently, for the same dog as in Part 2. This also examined many of the same factors relating to management covered in Part 2 and some of the equivalent data were compared to describe changes that had occurred during the lockdown period (change metrics Table 2).

Part 3 of the survey also evaluated the current status of a range of potential stress-related and problematic behaviours associated with someone trying to leave the room where the dog was. The following item stem “Since the lockdown began, has your dog started or increased doing any of the following behaviours?” was used to evaluate the emergence of the behaviours listed in Table 3, in the section “Problematic behaviours occurring during lockdown”. The answer choices were “Does not do this behaviour, Started, Increased, Decreased, Does this behaviour but no change”. Separation-related behaviours were not specifically asked about at this time, since it was anticipated that if the owners were working from home, the dogs would not be left alone frequently or for long periods.

A follow-up survey was sent automatically each month to all participants to explore how the dogs’ behaviour changed over time. This survey was also in three parts: Part 1 asked the owner to confirm their identity and that the dog was still with them, then asked whether their routines had changed and if so, how. Then, Parts 2 and 3 were as per the initial survey but used the last 4 weeks as the point of reference. Further follow-ups were sent each month to those who had given informed consent to the previous month’s survey, with this process repeated for a period of up to 8 months following the initial survey. Respondents were recruited globally over social media between May 2020 and July 2021.

### 2.2. Analysis for Aim 1: Effects of Pre-Existing Conditions, Management, and Demographics on Behaviour

#### 2.2.1. Bivariate Associations

In order to address our first aim to investigate the effect of pre-existing conditions, demographics, and management techniques, we used data from the first survey only. We initially generated simple descriptives and made a limited number of bivariate comparisons.

Initially, we split the dogs into two groups: those dogs who never or rarely showed SRBs and dogs who did show pre-existing SRBs prior to COVID (Part 2 data), defined as occurring at least quite frequently to always occurring when left alone. We further subset the data into (a) single-dog households and (b) all households. Then, we performed a chi-squared test to examine the differences between those dogs that were left alone for different lengths of time each day, for both (a) single-dog households and (b) all households. Thus, the chi-squared tests compared the number of dogs who did not perform the behaviour, or only did so rarely, against those who performed it at least quite frequently, against the time left alone of never/less than an hour, 1–3.5 h, 3.5–7 h, and more than 7.5 h for the two demographic groups.

#### 2.2.2. Bivariate Analysis of Pre-Existing Conditions and Change in the Number of Reported Behaviour Issues during Lockdown

We also examined whether the occurrence of individual pre-existing signs of separation-related problems (Part 2 data) was related to the number of behaviour issues changing during lockdown at the time of the survey. The latter was calculated from the behaviour issues reported in Part 3. To do this, we divided the population into two subpopulations that were assessed separately: those reporting an increase in behaviour issues during lockdown and those reporting a decrease at this time. We then used a Mann–Whitney test to compare the number of behaviour issues that had either increased or decreased, respectively, during lockdown (Part 3 data) with those with and without the pre-existing signs (Part 2 data) for each sub-population.

#### 2.2.3. Multivariable Analysis of Variables Related to SRBs

We built generalised linear models with both the eight separation-related behaviours when left without human contact and the variables in Table 1 included as independent variables for models for each of the 12 analysable problematic behaviours recorded in Part 3 of the survey, which were scored on a six-point scale ranging from never to every time. Four further models were built using composite dependent variables that grouped some of these individual items into higher level categories: vocalisation when family left the room (i.e., barking, growling, howling, and whimpering scores), contact seeking (i.e., shadowing; pressing body; and asking for attention scores), and potential stress-related behaviours (i.e., lip licking, etc.; stretching, scratching, or licking/chewing themselves excessively; “Shaking off”; yawning/nose licking; and blinking scores).

A multinomial probability distribution with cumulative logit link function was used in all models. Given the method of sampling single dogs from multi-dog households where management varied between dogs (dog’s name who came first in the alphabet) to generate specific dog demographics, we could not include dog demographic features in these models, so there were 15 independent variables in these models. Given the potential importance of specific dog demographics, we therefore ran a second set of models using just the data from single-dog households, which allowed us to include the age (as categories: 0–2 years old, 3–7 years old, 8+ years old), and sex (including neuter status) of the focal dog, with multi-dog household removed, leading to models with 16 independent variables. We did not examine the effect of breed because only one breed (Labradors) had more than 50 dogs and the next most common breed (German Shepherds) only had 22. To separate the datasets, the term “all-dog households” is used below for the full set of 1106 surveys, “multi-dog” for those dogs living with at least one other dogs, and “single-dog” for those living without other dogs.

### 2.3. Analysis for Aim 2: Exploring Behaviour Changes over Time

In order to address our second aim, we identified 104 respondents whom we could track from baseline for a further four months using the monthly follow-up surveys repeating the questions from Part 3. None of the dogs were being fostered in the home and none spent most of their time outside. We then used survival analysis to compare the emergence of the following problem behaviours during lockdown: vocalisation, contact seeking (i.e., the grouped behaviour of shadowing; pressing Body; and asking for attention), destruction of personal items, elimination (urination or defecation), and potential stress-related behaviours (defined as above in relation to our first aim). For those working from home, we calculated the length of time they had been working from home at each of the five sample points with the first day taken as the baseline, whereas for those continuing to work outside of the home we used the dates of the survey responses to determine intervals, with the first survey date used as the baseline. If someone changed from working from home to working outside of the home, they were removed from the database at that point. To explore the emergence of the problem behaviours, we excluded dogs who already performed the behaviours pre-COVID.

In order to examine the effect of time left alone on the emergence of problem behaviour, we divided the dataset on the basis of time left alone on a daily basis (once using a 1 h threshold and once using a 4 h threshold) and repeated a similar analysis but using the survey number as our measure of time.

## 3. Results

In total, 2387 respondents gave informed consent for the use of data from completion of the first survey, 722 did the same for month 1, 318 for month 2, 183 for month 3, 136 for month 4, 102 for month 5, 80 for month 6, 60 for month 7, and 46 for month 8.

### 3.1. Aim 1: Examining the Effect of Pre-Existing Conditions, Changes to Some Pre-Existing Management-Related Variables, and Dog Demographics on Behaviour Issues with Dogs during Lockdown

#### 3.1.1. Population Characteristics

After cleaning the data of incomplete responses, etc., 1106 respondents remained in the dataset from the first survey for analysis. There were 676 single-dog households, and 430 multi-dog households (range 2–17 dogs, mean = 2.52, standard deviation = 1.309, median = 2). Among the single-dog households, there were 329 females (22 intact, 307 spayed) and 347 males (36 intact, 311 neutered) from 78 breeds. The average age of these dogs was 6.2 +/- 3.6 years (mean +/- SD). In total, 375/1106 (33.9%) dogs showed at least one of the eight separation-related behaviours at least quite frequently. The distribution of the signs and their frequency are shown in Table 3.

#### 3.1.2. Relationship between Time Left Alone and Individual Pre-COVID Occurrence of Separation-Related Behaviours

In single-dog households, 105 dogs (15.5%) were left alone for less than an hour or never left alone, 186 dogs (27.5%) were left alone for 1 to 3.5 h, 220 dogs (32.5%) were left alone for 3.5–7 h, and 165 dogs (24.4%) were left alone for 7.5 h or more. For all-dog households, the ratios were similar with 168 dogs (15.2%), 319 (28.8%), 361 (32.6%), and 258 (23.3%), respectively.

Chi-squared test results revealed that dogs left alone for less than an hour were more likely to vocalise at least “quite frequently” when left. Dogs left alone for more than 3.5 h but fewer than 7 h were less likely to show destruction to and/or around windows and doors when left at least “quite frequently”, whereas dogs left alone for more than 7 h were more likely to show this level and type of destructiveness. No significant associations were found between being left for 1–3.5 h and behaviour when left.

#### 3.1.3. Relationship between Pre-Existing Separation-Related Behaviours and Change in Number of Potential Behaviour Issues during Lockdown

The data relating to those reporting an increase in the number of behaviour issues during lockdown and those reporting a decrease at this time (both recorded in Part 3 of the survey) were separated and the results for each subpopulation are summarised in Table 4.

Amongst those reporting an increase in the number of issues occurring during lockdown, there was a significantly greater number of issues increasing amongst those showing any of the pre-existing potential signs of separation-related problems. In other words, for those dogs whose behaviour deteriorated during lockdown, the occurrence of any pre-existing potential separation-related behaviour was associated with a greater increase in the number of behaviour issues.

Amongst those reporting a decrease in the number of issues occurring during lockdown, the results were more varied. Those showing signs of drooling, destruction of windows, or self-injury had a smaller reduction in the number of behavioural issues than those without these pre-existing signs. By contrast, those showing destruction of personal items, house soiling, or chewing if gated/crated had a greater reduction in the number of problem behaviours. It is worth noting that a numerically higher proportion of owners with dogs with pre-existing separation issues reported a decrease in most instances, the only exception being those who owned a dog with pre-existing drooling.

#### 3.1.4. Significant Influences on Problem Behaviour Issues at Home during the Pandemic

The prevalence of different levels of change in behavioural issues during lockdown used in the analyses are given in Table 5. These were the dependent variables used in the generalised linear models. Some of the models did not converge, and in these cases the factor which provided perfect prediction was identified and models were rerun without it. Pre-existing potential signs of separation-related problems were generally important in these models, but management was generally less significant and dog demographics were rarely of significance. The main results of each model are described in each subsection, with full details of the models provided in the Appendix A.

##### Vocalising When Family Members Leave the Room

Significant factors in the model based on all-dog households all related to pre-COVID separation-related behaviour signs and were vocalisation (*p* = 0.001), destruction to and/or around windows and doors (*p* = 0.019), and chewing on crate bars if crated/gated (*p* = 0.042).

Significant factors in the model based on single-dog households were also only related to pre-COVID separation-related behaviour signs: elimination (*p* = 0.024), destruction to and/or around windows and doors (*p* = 0.034), destruction of personal items (*p* = 0.025), and vocalisation (*p* = 0.038).

##### Barking When Family Members Leave the Room

Significant factors in the model based on all-dog households were vocalisation (*p* < 0.001), destruction of personal items (*p* = 0.035), chewing on crate bars if crated/gated (*p* < 0.001), and change in safe space (*p* = 0.006).

The significant factors in the model based on single-dog households were vocalisation (*p* = 0.009), destruction of personal items (*p* = 0.007), change in time alone (*p* = 0.041), change in safe space provision (*p* = 0.049), and age category (*p* = 0.016).

##### Growling When Family Members Leave the Room

Significant factors in the model based on all-dog households were chewing on crate bars (*p* = 0.008) and change in the safe space (*p* = 0.022).

The significant factors in the model based on single-dog households were change in safe space (*p* = 0.003) and change in social play (*p* = 0.032).

##### Whimpering/Whining When Family Members Leave the Room

There were no significant factors in the final model based on all-dog households.

When vocalisation was included in the model based on single-dog households with the other variables, non-convergence occurred, suggesting this was a perfect predictor. A separate model without this factor allowed convergence with the other variables and indicated that the age category (*p* = 0.048) and destruction to and/or around windows and doors (*p* = 0.012) were significant.

##### Contact Seeking

Significant factors in the model based on all-dog households were change in time alone (*p* < 0.001) and change in number of walks (*p* = 0.006).

The only significant factor in the model based on single-dog households was change in time alone which did not cause non-convergence, thus was not a perfect predictor, while explaining significant variation in the model (*p* = 0.016).

##### Shadowing/Following Family Members from Room to Room

Significant factors in the model based on all-dog households were vocalisation (*p* = 0.036) and someone else home (*p* = 0.041).

The only significant factor in the model based on single-dog households was self-injury (*p* = 0.041).

##### Pressing Body against or Sitting on Family Members

Significant factors in the model based on all-dog households were vocalising (*p* = 0.040), someone else being home (0.039).

Significant factors in the model based on single-dog households were change in time alone (*p* = 0.048) and change in number of walks (*p* = 0.037), while vocalising was significant at *p* = 0.050.

##### Attention Seeking, i.e., Asking for Attention or to Play More Frequently (Barking at Family Members, Whimpering, Mouthing, Nudging, Scratching on the Door to Room Where a Person Is)

Significant factors in the model based on all-dog households were change in time alone (*p* = 0.029), change in social play (*p* = 0.043), change in number of walks (*p* = 0.002), running around to/from windows and doors (*p* = 0.045), and chewing crate bars (*p* = 0.009).

The significant factors in the model based on single-dog households were chewing on crate bars (*p* = 0.006), change in time alone (*p* = 0.030), and sex and neuter status (*p* = 0.013).

##### Blocking Access—Trying to Stop Family Members from Leaving the House or Room, for Example by Standing in Front of the Door Barking

The only significant factor in the model based on all-dog households was drooling (*p* = 0.015), with change in time alone significant at *p* = 0.050.

The only significant factor in the model based on single-dog households was sex and neuter status (*p* = 0.014), with chewing crate bars significant at *p* = 0.050 as males blocked more than females.

##### Potential Signs of Stress (Grouped Variable)

Significant factors in the model based on all-dog households were multi-dog household (*p* = 0.002), running to and/or around windows and doors (*p* = 0.035), elf-injury (*p* = 0.015), change in number of walks (*p* = 0.029), and change in provision of sound (*p* = 0.019).

Significant factors in the model based on single-dog households were self-injury (*p* = 0.038) and change in provision of sound (*p* = 0.002).

Increasing Behaviours such as Stretching, Scratching, or Licking/Chewing Themselves Excessively

Significant factors in the model based on all-dog households were multi-dog household (*p* < 0.001), drooling (*p* = 0.033), self-injury (*p* < 0.001), and change in number of walks (*p* = 0.004).

Significant factors in the model based on single-dog households were change in number of walks (*p* = 0.002), chewing on crate bars (*p* = 0.037), self-Injury (*p* < 0.001), and sex and neuter status (*p* = 0.036).

##### “Shaking Off” as If Wet

Significant factors in the model based on all-dog households were multi-dog household (*p* < 0.001) and running around to/from windows and doors (*p* = 0.012).

Significant factors in the model based on single-dog households were a change in number of walks (*p* = 0.045), while drooling approached significance (*p* = 0.051).

##### Yawning/Nose Licking

Significant factors in the model based on all-dog households were running to and/or from windows and doors (*p* = 0.039), self-injury (*p* = 0.013), if someone else was home (*p* = 0.037), change in provision of sound (*p* = 0.023), and multi-dog household (*p* < 0.001).

Significant factors in the model based on single-dog households were destruction of personal items (*p* = 0.017), change in provision of sound (*p* = 0.008), self-Injury (*p* = 0.015), and if someone else was home (*p* = 0.045).

##### Blinking

Significant factors in the model based on all-dog households were self-injury (*p* = 0.036) and change in provision of sound (*p* = 0.026).

The only significant factor in the model based on single-dog households was destruction of personal items (*p* = 0.018).

##### Destruction of Personal Items Such as Couch, Pillows, Book, Shoes, Hats, etc.

Significant factors in the model based on all-dog households were drooling (*p* = 0.028), running around to/from windows and doors (*p* = 0.017), destruction to and/or around windows and doors (*p* = 0.010), and if someone else was home (*p* <0.001). Notably, destruction of personal items when left alone pre-COVID was not a significant factor (*p* = 0.443).

Significant factors in the model based on single-dog households were vocalising (*p* = 0.041), drooling (*p* = 0.042), destruction to and/or around windows and doors (*p* = 0.040), destruction of personal items (*p* < 0.001), chewing crate bars (*p* < 0.001), self-injury (*p* = 0.029, someone else being home (*p* <0.001), and change in time alone (*p* = 0.044).

##### Social Breakdown/Snarling at Owner or Other Dog

Significant factors in the model based on all-dog households were destruction to and/or around windows and doors (*p* = 0.031), running around to/from windows and doors (*p* = 0.033), and multi-dog household (*p* = 0.046).

There were no significant factors in the model based on single-dog households, possibly due to the small number of dogs that produced the behaviour (15.7% of sampled dogs).

##### Repetitive Behaviours

Including the factor destruction to personal items caused non-convergence in the all-dog household dataset, suggesting it was a perfect predictor. Removing it created a model where only multi-dog household was significant (*p* = 0.019).

Significant factors in the model based on single-dog households were destruction to personal items (*p* = 0.042), chewing crate bars (*p* = 0.008), and age category (*p* = 0.008).

### 3.2. Aim 2: Examining the Emergence of Potentially Problematic Behaviour in Dogs over Time during the Pandemic

Within this population, 104 respondents provided data for analysis: 70 were single-dog households and 34 multi-dog households (24 with 2 dogs, 8 with three dogs and one each with 4 and 5 dogs). Of the single dogs sampled, 32 were purebred (from 23 different breeds), 45 were multiple crossed breeds, and 27 were a single crossbreed. There was one entire bitch, 55 neutered females, 44 neutered males, and 4 entire males.

The prevalence of behaviours of interest amongst those who continued to work outside of the home versus those who were working from home is provided in Table 6.

After excluding subjects who reported the behaviour of interest occurring at baseline, survival analysis comparing the emergence of these problem behaviours during lockdown could be undertaken for the following: vocalising (as defined above, grouping barking, growling, howling, and whimpering/whining), contact seeking (grouping shadowing/following family members from room to room; pressing body against or sitting on family members; attention-seeking behaviours such as whimpering), social breakdown (i.e., snarling at owner), blocking access, repetitive behaviour, elimination, and stress signs (i.e., yawning and blinking).

This revealed significant differences in the emergence of social breakdown (Chi-sq Breslow = 6.037, d.f. = 1, *p* = 0.014), which appeared consistently to affect about 10–15% of subjects who worked outside the home, whereas for those working from home there was an initially lower risk of this but a cumulative increase over time (Table 6), with approximately twice the risk by month 4 of the survey. Further analysis indicated that males drove this difference (Chi-sq Breslow = 8.722, d.f. = 1, *p* = 0.003), with no significant difference for working patterns for female dogs (Chi-sq Breslow = 0.572, d.f. = 1, *p* = 0.449).

When we repeated the analysis using time left alone as either 4 h or 1 h to separate the groups, no differences occurred between the two groups across the five surveys.

## 4. Discussion

Pre-COVID, approximately 20% of dogs in our sample showed signs of separation-related vocalisation at least quite frequently when left alone for more than an hour and 15% of dogs showed running at windows, but only about 5% of dogs showed some form of destructiveness or elimination in these circumstances. COVID lockdowns generally increased the prevalence of dogs presenting separation-related problem behaviours (Table 5) and this increase was generally more often significant in dogs with pre-existing problem behaviours, though some dogs did show a significant decrease in destructive behaviours compared to baseline.

In general, dogs with pre-existing signs of separation-related problems were more likely to express behavioural issues and to have more issues than dogs without pre-existing problems (Table 4). In this regard, pre-existing vocalisation seemed to be particularly important as it was associated with all four of the behaviours associated with separation (leaving a room) during lockdown as well as shadowing the owner around the home and pressing body against family members for the all-dogs sample. Similarly, chewing at crate bars, etc., when confined was predictive of three of the four separation-related problems during lockdown (not whining/whimpering), which could all be explained by poor frustration tolerance [20,21,22,23]. Destructiveness of and running at windows were also important predictive pre-COVID signs, as they were significantly related to whining, social breakdown, and a range of potential signs of stress. Although whining may change in form as separation time increases [24], these results indicate that this is perhaps not so much an expression of the strength of their social bond with the owner but rather perhaps an expression of social frustration. 

The amount of time left alone was also important. Dogs appeared more likely to vocalise and be destructive if left for between 3.5 and 7 h and to continue to be destructive if left for more than 7 h. This is consistent with ethological work showing a shift in behavioural tendencies associated with increased distress over time [25,26] and recent epidemiological work which found four general strategies pursued by dogs with these problems which appear to be largely associated with frustration [20]. Overall, more dogs decreased their destruction of personal items than increased (Table 5), but this may be because dogs were left less frequently and for shorter periods of time compared to pre-COVID, limiting their opportunities for destructive behaviour.

Vocalisation pre-COVID was important in predicting several social behaviours during lockdown, including vocalising, shadowing, and pressing the body against the owner, but not to the stress behaviours including destruction, blocking access, social breakdown, or self-injury. Interestingly, self-injurious behaviour pre-COVID was frequently a significant predictor and was predictive of several related behavioural issues during lockdown that were not predicted by vocalising: destroying personal items, blocking attempts at separation, and multiple stress-related behaviours, including stretching and yawning, as well as shadowing, which was predicted by vocalising. Thus, yawning and stretching may be indicative of social stress [27], although they are different to the signs reported to occur in inter-dog interactions [28]. This potentially indicates a relationship between social stress and self-mutilation, potentially as a form of redirected aggression [29] as has been reported in humans [30,31]. The association with yawning is particularly intriguing given its potential relationships with social cognition, e.g., a shared contagion suggesting empathy [32]. In contrast to pre-existing separation-related problems, most changes in management were not associated with specific separation-related issues during COVID lockdown, e.g., change in social play opportunities was only significant as a predictor in two of the models. A change in safe space provision was associated with increased barking and growling when the owner tried to leave a room, but not other vocalisations when the owner tried to leave a room. This might reflect a reduced sense of security or a change in the social dynamic, resulting in the dog potentially trying to increase control over their environment, which would be expected when the safe space is a place where dogs feel in control and thus unthreatened [33]. Increased contact seeking by dogs was predicted by disruption to social routines, such as time alone and changes to walking behaviour, which might reflect the importance of the owner as a social buffer [34], at least when the relationship is of good quality [35]. Changes in time alone during the COVID lockdown seemed to also result in wider changes in social behaviour, and more often in single-dog households, possibly reflecting the lack of other social contact. Effects were seen on following behaviour around the home, contact seeking, and destruction of personal items. Furthermore, the pre-COVID behaviour of attempting to escape crate/chew crate bars was more often predictive in single-dog household models, where it was significant for stretching, access blocking, and asking for attention, whereas in multi-dog households it was only important in vocal behaviour models. These results all reinforce previous concerns relating to changes that the pandemic had on the social behaviour of dogs [4,10,15,16], but highlight the complexity of the dynamics involved and a need for caution in simple generalisations.

Concerns over the impact of changes in exercise routine [4,9] also appear justified, with our data finding that changes in the number of walks were associated with increased contact seeking, attention seeking, stretching, and shaking off. The first two of these could predispose the dog to a range of attention-seeking behaviours that might impact the dog–owner relationship [35]. Interestingly, the former behaviours have been at least anecdotally associated with musculoskeletal discomfort in dogs [36], and this might also partly explain the relationship between this behaviour and age in our data. Whole-body “wet dog shakes” may also be associated with improving lymphatic flow [37], which ordinarily might be expected to occur with exercise. Thus, these behaviour changes might reflect compensatory activity for a reduction in exercise. In addition, whole body shakes in rats may be a correlate of central serotonin activity [38], which has been associated with greater inhibitory control in dogs [39]. Therefore, this finding could also suggest that these responses are indicative of some form of attempt to cope with reduced exercise through a greater inhibition of behaviour in general. These hypotheses deserve further research given their potential role in self-regulation and our generally poor understanding of the function of specific behaviours [40].

Although we did not investigate many demographic factors, it is noticeable how some, such as sex and neuter status, were predictive of only a small number of behaviours, e.g., attention seeking, access blocking, and stretching during lockdown in our models, and not the emergence of signs indicative of conflict or aggression, which the public often relate to sex and neuter status. We found that male dogs were more likely to block their owners from leaving the room, but whether this is indicative of an aggressive act, a more male-typical behaviour, or related to a separation-related issue is unclear. In this regard, it is worth noting that the increase in social breakdown over time seemed to be driven as an effect largely by male dogs. By contrast, work on risk factors for separation-related problems has reported inconsistent gender and neuter effects [41], with some contradictory findings, e.g., Flannigan and Dodman [42] found that sexually intact dogs were less likely to have these problems [42], while McGreevy and Masters [43] found that intact dogs had a higher probability of exhibiting high separation-related behaviour scores [43]. The latter study also reported that dogs who engaged in games with their owners had lower scores, but we did not find an association with potential separation-related issues and a reduction in play. Rather, the latter was associated more with attention seeking and growling, potentially reflecting the powerful reinforcing value of play (with many dogs growling in play). Play is an important social activity and being part of a multi-dog household might be expected to be associated with a lower risk of issues related to the social or exercise needs of dogs if playing with other dogs compensates for a reduction in other forms of exercise during lockdown. In our data, being in a multi-dog household was associated with decreases in stretching and whole-body shakes, which might relate to the exercise function of these behaviours but also changes in a range of potential stress-related behaviours and the risk of social breakdown between the dog and owner. It is also notable that overall the risk factors for behaviour issues during lockdown showed quite a lot of difference between single- and multi-dog households, highlighting the multifactorial and heterogeneous nature of this problem [20] and the need for caution in focusing on simple causal explanations to such complex issues [44].

Although a change to time left alone was associated with changes in a number of behaviours associated with close social contact (attention seeking and body contact, etc.), it was not strongly associated with potential separation-related issues. Actual time spent alone was perhaps less important than we might predict as it had no predictive value in the emergence of problem behaviour during the 4 months that we reviewed. However, our initial data suggested that perhaps the most prolonged periods of separation may be the most important. The changes to time alone were associated with changes in physical contact with the owner and attention seeking by dogs in both single- and multi-dog households, supporting the importance of physical contact in close human–dog relationships [45,46] and the differential role of the human over another dog in the attachment-related needs of a dog [47]. Our longitudinal analysis of risk of behavioural issues during lockdown does not support the view that a change in working from home, per se, is a strong driver for the development of behaviour problems. Instead, the preceding data analysis indicates that the human response to this might be an important factor in the development of these issues. This is supported by the longitudinal finding that among owners who worked from home, the risk of a breakdown in social relationships between the dog and owner was delayed and accelerated over time. This, and other numerical trends observed over time (Table 6), indicate a possibly cumulative effect over time and a potentially strong effect of inappropriate reinforcement of contact-related behaviour over time. Given the relatively small sample size for these models, these findings should be considered tentative, but they offer a potentially intriguing insight into the dynamic of the human–animal relationship during this time. Such behaviour is also consistent with the widespread decline in human health reported over the lockdown period, with the owners who suffered the most themselves also reporting more severe problems with their pet [15]. We therefore suggest that the content and consistency of interactions between owners and their dogs during lockdown may be more important than dog-related factors, outside of pre-existing separation-related problems. This result fits with a previous report [43] stating that 16% of owners could recall a change in the home, including a change in work routine, associated with the onset of separation-related problems, whereas only 10% of owners of dogs who had other problems could make such an association, which suggested a two-fold increased risk in relation to separation-related problems. This finding is consistent with both our result and the only other longitudinal study undertaken during the COVID lockdown to date [17], which also identified the importance of pre-existing separation-related problems. Thus, we suggest that these dogs were at risk of developing or intensifying a wider range of behavioural issues during lockdown. The patterns of associations that emerge are also consistent with the growing view that separation-related problems are not a reflection of some form of hyper-attachment [48], as has been previously suggested [49], even if the signs may be interpreted as thus. Instead, separation-related problems may be related to the quality/form of the attachment bond that exists between the dog and owner [50], and particularly the role of this in the development of a predisposition towards expressions of frustration by the dog [20,23,51].

## 5. Conclusions

Our results provide important insights into dog behaviour and its adaptation to changing circumstances, especially those affecting owners. In particular the role of frustration and inadvertent forms of social reinforcement (both consistent and inconsistent) may be key to the development of many issues. Separation-related problems appear to be an important risk factor affecting a dog’s ability to cope with other stressors. Different signs of separation-related problems predispose dogs to the development of different potential issues when the owner is at home, and so it is important to understand the specific functional role of the individual signs. Separation-related vocalisation, chewing to escape confinement, and self-injury seem to particularly increase the risk of other signs of distress related to owner activity that might lead them to be out of the sight of the dog, such as barking when left alone, seeking attention from the owner by pressing their body against them or shadowing them, and stretching or chewing on themselves. The need for social contact seemed to be increased when there were changes in the dog’s routines, such as its exercise schedule or time spent alone, but not by the dog’s sex or age. A reduction in exercise was also related to what might be potential compensatory efforts at muscular activity in the form of increased stretching and body shaking. Changes to the safe space of the dog were related to a number of behavioural changes, which seemed to reflect an increased need for control over the environment. There was evidence of an increased risk of social breakdown over time amongst owners who worked from home, that was not apparent amongst those working outside the home. This might reflect the cumulative stress of the pandemic on dogs but also its impact on the health of their human carers.

## Figures and Tables

**Table 1 vetsci-10-00195-t001:** Problematic behaviours as observed by their owners and the wording used in the survey to describe them. A six-point ordinal scale was used for both parts. For Part 2, as it related to pre-existing behaviours, the question was “When left without human company prior to the lockdown, how often did your dog show the following behaviours if left for more than 1 h” with a six-point ordinal rating scale relating to frequency in the given context (every time, most of the time, about half the time, less than half the time but quite frequently, rarely, never). For Part 3, the question was phrased as “Since the lockdown began, has your dog started or increased doing any of the following behaviours?” and the six-point ordinal scale options were “Does not do this behaviour, Started, Increased, Decreased, Does this behaviour but no change”.

Survey Part	Behaviour	Simplified Term
Part 2—Pre-existing separation-related problematic behaviours	Vocalising when family members leave the room	Vocalisation
	Excessive drooling/salivating	Drooling
	Running around to/from windows	-
	Destruction to and/or around windows and doors	-
	Destruction of personal items such as couch, pillows, book, shoes, hats, etc.;	Personal item destruction
	Elimination	-
	If crated or gated, chewing bars/escaping	Chewing crate bars
	Self-injury, e.g., from excessive licking	Self-injury
Part 3—Problematic behaviours occurring during lockdown	Barking when family members leave the room	Barking
	Growling when family members leave the room	Growling
	Howling when family members leave the room (this was subsequently excluded from analysis as it was not reported to occur)	Howling
	Whimpering/whining when family members leave the room	Whimpering/whining
	Shadowing/following family members from room to room	Shadowing
	Pressing body against or sitting on family members	Pressing body
	Asking for attention or to play more frequently (barking at family members, whimpering, mouthing, nudging, scratching on the door to room where a person is)	Attention-seeking
	Growling, snarling, lunging, biting, or other signs of confrontation towards family members or other animals	Social breakdown
	Trying to stop family members from leaving the house or room, for example by standing in front of the door barking	Blocking access
	Unusual and/or repetitive behaviours that do not seem to make any sense. For example, shadow fixating, spinning in a circle (repetitive behaviours), or chasing tails (this was subsequently excluded from analysis as it was not reported to occur)	Repetitive behaviour
	Changes in behaviour when interacting with family members, such as licking lips, yawning more than usual, drooling, lifting a paw, crouching or cringing, or becoming tense and stiff (this was subsequently excluded from analysis as it was not reported to occur)	Lip licking
	Increasing behaviours such as stretching, scratching, or licking/chewing themselves excessively	Stretching
	“Shaking off” as if wet	Shaking off
	Destruction of personal items such as couch, pillows, book, shoes, hats, etc.	Destruction of personal items
	Yawning/nose licking	Yawning

**Table 2 vetsci-10-00195-t002:** Management variables used in analysis. Fixed metrics relate to baseline variables assessed pre-COVID taken from Part 2 of the survey; change metrics were extracted from Parts 2 and 3 of the survey to determine changes pre- versus post-COVID which might affect dog behaviour. Simplified terms refer to text used to describe the metric for narrative purposes henceforth.

Item	Rating	Fixed Metric Used in Analysis	Change Metric Used in Analysis	Simplified Term
Please state the number of times a week your dog played with dogs from outside the home.	Frequency	-	Increase, decrease, no change	Change in social play
How many walks did your dog(s) typically have each weekday and for how long? Up to 10 walks specified	Frequency	-	Increase, decrease, no change	Change in number of walks
Did your dog have its own special place to go to in the home where they would not be disturbed whether home alone or not?	Binary: Yes/No	-	Provision, loss, no change	Change to safe space provision
How long was your dog left without access to human company each day?	6-point scale of hourly ranges	-	Increase, decrease, no change in time	Change in time left alone
How many dogs?	Count	Multi-dog household or not	-	Multi-dog household
Were any of the behaviours less likely if someone other than you was in the home?	No, anyone Yes, only specific people	Categories	-	Behaviour if someone home
Did you typically play sounds when leaving your dog without human company?	7 options of sound including none	Categories	Provision, loss, no change	Provision of sound

**Table 3 vetsci-10-00195-t003:** Prevalence of different frequencies of 8 separation-related behaviours when left without human company for more than 1 h (potential signs of a separation-related behaviour problem) prior to the lockdown, in the sample of all-dog households, * “Never” responses for this item include dogs who are never crated.

Behaviour	Every Time		Most of the Time		About Half the Time		Less Than Half the Time but Quite Frequently		Rarely		Never	
	N	%	N	%	N	%	N	%	N	%	N	%
Vocalising when family members leave the room	53	4.8	53	4.8	40	3.6	70	6.3	353	31.9	537	48.6
Excessive drooling/salivating	6	0.5	7	0.6	4	0.4	9	0.8	110	9.9	970	87.7
Running around to/from windows	28	2.5	39	3.5	38	3.4	76	6.9	189	17.1	736	66.5
Destruction to and/or around windows and doors	3	0.3	6	0.5	6	0.5	27	2.4	78	7.1	986	89.2
Destruction of personal items such as couch, pillows, book, shoes, hats, etc.	2	0.2	10	0.9	10	0.9	37	3.3	190	17.2	857	77.5
Elimination	3	0.3	15	1.4	17	1.5	36	3.3	195	17.6	840	75.9
If crated or gated, chewing bars/escaping *	9	0.8	12	1.1	7	0.6	9	0.8	48	4.3	1021	92.3
Self injury, e.g., from excessive licking	1	0.1	7	0.6	3	0.3	24	2.2	66	6	1005	90.9

**Table 4 vetsci-10-00195-t004:** Number and proportion of subjects with pre-existing potential signs of separation-related problems showing an increase or decrease in the number of behavioural issues during lockdown and the median change in number of issues in each group. Raw data figures relate to the number with the relevant population (e.g., 58 dogs showed an increase in behaviour issues during lockdown out of 87 who carried out “Vocalisation”). A Mann–Whitney test was used to compare differences in the change in number of behaviour issues between those with and without the pre-existing potential separation-related problem sign, for each of the two subpopulations.

Pre-Existing Potential Sign of Separation-Related Behaviour Problem	Dogs Increasing the Number of Behaviour Issues at the Time of Survey, amongst Those with		Dogs Decreasing the Number of Behaviour Issues at the Time of Survey, amongst Those with	
	Occurrence of Pre-Existing Behaviour Issue	Absence of Pre-Existing Behaviour Issue	Occurrence of Pre-Existing Behaviour Issue	Absence of Pre-Existing Behaviour Issue
**Vocalisation** % change Median change in no. of issues Test statistic and *p*-value	58/87	187/385	11/87	43/385
	66.7	48.6	12.6	11.2
	3	2	1	1
	32,666; *p* < 0.001		3204; *p* = 0.061	
**Excessive drooling** % change Median change in no. of issues Test statistic and *p*-value	4/7	241/465	0/7	54/465
	57.1	51.9	0	11.6
	2	2	0	1
	30,159; *p* < 0.001		1709.5; *p* < 0.001	
**Running at windows/doors** % change Median change in no. of issues Test statistic and *p*-value	53/80	190/392	12/80	42/392
	66.3	48.5	15	10.7
	3	2	1	1
	32,416; *p* < 0.001		3057.5; *p* = 0.418	
**Destruction of windows/doors** % change Median change in no. of issues Test statistic and *p*-value	11/17	233/455	9/17	45/455
	64.7	51.2	52.9	9.9
	4	2	1	1
	30,950; *p* < 0.001		1291; *p* < 0.001	
**Destruction of Personal items** % change Median change in no. of issues Test statistic and *p*-value	18/25	227/447	11/25	43/447
	72	50.8	44	9.6
	4	2	1.5	1
	31,230; *p* < 0.001		1565; *p* < 0.001	
**House soiling** % change Median change in no. of issues Test statistic and *p*-value	20/32	225/440	10/32	44/440
	62.5	51.1	31.3	10
	3.5	2	1.5	1
	31,969; *p* < 0.001		1838; *p* < 0.001	
**Chewing if gated/crated** % change Median change in no. of issues Test statistic and *p*-value	10/13	235/459	5/13	49/459
	76.9	51.2	38.5	10.7
	3.5	2	2	1
	30,361; *p* < 0.001		1648; *p* < 0.001	
**Self-injury** % change Median change in no. of issues Test statistic and *p*-value	7/9	238/463	5/9	49/463
	77.8	51.4	55.6	10.6
	2	2	1	1
	30,069; *p* < 0.001		1417; *p* < 0.001	

**Table 5 vetsci-10-00195-t005:** Prevalence of behaviour issues and categories of behaviour used as dependent variables. The term all-dog households refers to data from the full set of 1106 surveys, and “single-dog” households to those living without other dogs.

Behaviour Category	Dataset	Does Not Do the Behaviour		Started Doing the Behaviour		Did the Behaviour Already but No Change		Did the Behaviour Already and Increased		Did the Behaviour Already and Decreased	
		N	%	N	%	N	%	N	%	N	%
Vocalising (including barking, growling, or whimpering/whining) when family members leave the room	All-dog households	**807**	73.0	**72**	6.5	**96**	8.7	**87**	7.9	**44**	4.0
	Single-dog households	**491**	72.6	**46**	6.8	**58**	8.6	**56**	8.3	**25**	3.7
Barking	All-dog households	**925**	83.6	**29**	2.6	**73**	6.6	**41**	3.7	**38**	3.4
	Single-dog households	**562**	83.1	**17**	2.5	**50**	7.4	**27**	4.0	**20**	3.0
Growling	All-dog households	**1065**	96.3	**3**	0.3	**11**	1.0	**7**	0.6	**20**	1.8
	Single-dog households	**653**	96.6	**1**	0.1	**7**	1.0	**6**	0.9	**9**	1.3
Howling	All-dog households	**1029**	93.0	**18**	1.6	**23**	2.1	**12**	1.1	**24**	2.2
	Single-dog households	**634**	93.8	**12**	1.8	**13**	1.9	**6**	0.9	**11**	1.6
Whimpering/whining	All-dog households	**898**	81.2	**60**	5.4	**64**	5.8	**56**	5.1	**28**	2.5
	Single-dog households	**546**	80.8	**35**	5.2	**40**	5.9	**41**	6.1	**14**	2.1
**Contact seeking** (including shadowing/following family members from room to room; pressing body against or sitting on family members; attention-seeking behaviours such as whimpering)	All-dog households	**124**	11.2	**83**	7.5	**374**	33.8	**490**	44.3	**35**	3.2
	Single-dog households	**66**	9.8	**56**	8.3	**218**	32.2	**314**	46.4	**22**	3.3
Shadowing/following family members from room to room	All-dog households	**244**	22.1	**88**	8.0	**469**	42.4	**270**	24.4	**35**	3.2
	Single-dog households	**141**	20.9	**51**	7.5	**297**	43.9	**165**	24.4	**22**	3.3
Pressing body	All-dog households	**308**	27.8	**50**	4.5	**512**	46.3	**213**	19.3	**23**	2.1
	Single-dog households	**188**	27.8	**35**	5.2	**307**	45.4	**135**	20.0	**11**	1.6
Asking for attention or to play more frequently	All-dog households	**317**	28.7	**84**	7.6	**324**	29.3	**348**	31.5	**33**	3.0
	Single-dog households	**175**	25.9	**56**	8.3	**199**	29.4	**228**	33.7	**18**	2.7
Destroying personal objects (e.g., pillows, books, etc.)	All-dog households	**936**	84.6	**11**	1.0	**88**	8.0	**11**	1.0	**60**	5.4
	Single-dog households	**571**	84.5	**8**	1.2	**52**	7.7	**9**	1.3	**36**	5.3
Blocking access (e.g., trying to stop family members from leaving the house or room)	All-dog households	**988**	89.3	**44**	4.0	**37**	3.3	**33**	3.0	**4**	0.4
	Single-dog households	**601**	88.9	**27**	4.0	**23**	3.4	**24**	3.6	**1**	0.1
Social breakdown (i.e., growling, snapping, or snarling at owner)	All-dog households	**942**	85.2	**24**	2.2	**69**	6.2	**38**	3.4	**33**	3.0
	Single-dog households	**570**	84.3	**12**	1.8	**52**	7.7	**24**	3.6	**18**	2.7
Repetitive behaviour	All-dog households	**978**	88.4	**38**	3.4	**63**	5.7	**19**	1.7	**8**	0.7
	Single-dog households	**587**	86.6	**30**	4.4	**42**	6.2	**13**	1.94	**4**	0.6
**Potential signs of stress** (including increasing behaviours such as stretching, scratching, or licking/chewing themselves excessively; lip-licking; “Shaking off” as if wet; yawning/nose licking; blinking)	All-dog households	**448**	40.5	**38**	3.4	**448**	40.5	**152**	13.7	**20**	1.8
	Single-dog households	**251**	37.1	**26**	3.8	**277**	41.0	**110**	16.3	**12**	1.8
Stretching, scratching, licking self	All-dog households	**702**	63.5	**66**	6.0	**209**	18.9	**110**	9.9	**19**	1.7
	Single-dog households	**552**	81.7	**24**	3.6	**57**	8.4	**39**	5.8	**4**	0.6
Lip-licking	All-dog households	**924**	83.5	**41**	3.7	**80**	7.2	**54**	4.9	**7**	0.6
	Single-dog households	**397**	58.7	**46**	6.8	**141**	20.9	**82**	12.1	**10**	1.5
Shaking off	All-dog households	**672**	60.8	**35**	3.2	**354**	32.0	**39**	3.5	**6**	0.5
	Single-dog households	**469**	69.4	**11**	1.6	**185**	27.4	**7**	1.0	**4**	0.6
Yawning	All-dog households	**627**	56.7	**25**	2.3	**388**	35.1	**57**	5.2	**9**	0.8
	Single-dog households	**349**	51.6	**19**	2.8	**260**	38.5	**44**	6.5	**4**	0.6
Blinking	All-dog households	**779**	70.4	**15**	1.4	**294**	26.6	**10**	0.9	**8**	0.7
	Single-dog households	**469**	69.4	**11**	1.6	**185**	27.4	**7**	1.0	**4**	0.6

**Table 6 vetsci-10-00195-t006:** Occurrences of each problem behaviour during the five months surveyed. Note that as participants stopped working from home, they were removed from the group “Work from home”, therefore, the number of dogs included by month 4 was smaller than that in the baseline; percentages are calculated from the numbers participating in that month’s survey. Thus, while the number of dogs blocking access did not increase from month 2 to month 4 in the work from home group, the percentage overall did as the overall sample size fell.

	Total	Working Outside of the Home					Work from Home				
Survey Point	Behaviour Prior to COVID	Baseline	Mo 1	Mo 2	Mo 3	Mo 4	Baseline	Mo 1	Mo 2	Mo 3	Mo 4
Vocalising	54 (51.9%)	1 (7.7%)	1 (11.1%)	1 (16.7%)	1 (16.7%)	1 (16.7%)	1 (2.8%)	3 (10.0%)	3 (10.0%)	3 (16.7%)	5 (31.25%)
Contact seeking	N/A	19 (36.9%)	14 (92.6%)	9 (100%)	9 (100%)	8 (100%)	45 (52.9%)	63 (94.0%)	56 (100%)	46 (100%)	46 (100%)
Social breakdown	N/A	2 (11.8%)	2 (16.7%)	1 (12.5%)	1 (12.5%)	1 (14.3%)	2 (2.4%)	5 (7.5%)	10 (17.9%)	8 (17.4%)	9 (22.5%)
Blocking access	N/A	0 (0%)	0 (0%)	0 (0%)	0 (0%)	0 (0%)	6 (7.1%)	7 (10.4%)	8 (14.3%)	8 (17.4%)	8 (20.0%)
Repetitive behaviour	N/A	0 (0%)	0 (0%)	0 (0%)	1 (11.1%)	1 (12.5%)	4 (4.9%)	7 (11.7%)	11 (24.4%)	10 (27.8%)	11 (37.9%)
Elimination	28 (26.9%)	0 (0%)	0 (0%)	0 (0%)	0 (0%)	0 (0%)	0 (0%)	1 (1.7%)	2 (4.1%)	3 (7.5%)	3 (8.8%)
Destroying personal objects (pillows)	22 (21.2%)	0 (0%)	0 (0%)	0 (0%)	0 (0%)	1 (37.5%)	2 (3.1%)	0 (0%)	4 (10.3%)	5 (16.1%)	5 (20.0%)
Stress	21 (20.2%)	2 (15.4%)	5 (50%)	4 (57.1%)	4 (57.1%)	3 (50%)	17 (25.0%)	37 (67.3%)	34 (73.9%)	30 (83.3%)	26 (83.9%)

## Data Availability

Data are held by the University of Lincoln and anonymised records will be made available upon reasonable request.

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
