# Peer review of "Changes in Dog Behaviour Associated with the COVID-19 Lockdown, Pre-Existing Separation-Related Problems and Alterations in Owner Behaviour"

_vetsci, 2023, doi:10.3390/vetsci10030195_

Round 1

Reviewer 1 Report

This is a really nice sample that was conducted at numerous time points over lockdowns, reducing the risk of recollection bias. There is a lot of information within the manuscript which, at times, becomes a little opaque due to the writing style. I would encourage the authors to see where the text could be reduced to improve readability without losing the core meaning.

L21/36: the term "virtual absence" is odd, even with the e.g. I find it a little difficult to understand, possibly due to my own unfamiliarity. Is it "intended absence"?

L23: dog's

L65: maybe (or just may)

L73: worsen rather than deteriorate

L83-93: This is a rather lengthy report of a single paper, could it be better integrated within the intro? It would also be good to see a general intro to separation issues and their impact on owners and dogs. It needn't be exhaustive.

L85-87: a little confusing as written. "However, of dogs with pre-existing separation-related behaviour problems, approximately 50% ceased to display them during lockdown" (I assume because the triggers were absent/reduced?).

L97: dataset

L97-99: this is a long sentence, I would suggest breaking it up for clarity. ("over time. This was based" "during the pandemic. Especially so given"). I would delete "because of the pandemic",  it is tautological.

In M&Ms questionnaire part 2 is it possible to make reference to the behaviours outlined in Table 2 rather than listing them? Otherwise, the text is quite dense. (e.g. L106 The occurrence of 8 behaviours (see table n) were rated... etc.)

L125: Similarly, could the 16 behaviours be referred to a table (in this case 4)? Retaining those that were removed from analysis through non-occurrence in text.

L148-188: The analyses are rather hard to follow, is it possible to break them down, possibly by simply bullet-pointing each step. Long paragraphs of multiple steps can be quite confusing.

I note that the sample was global. How was this addressed based on international differences in lockdown across the 8 months? Was location included in the analyses as a variable? This is perhaps a substantial limitation and the data on respondents should provide this info, assuming respondent data were collected.

L 223-226: The writing in these sections is rather convoluted and so the meaning is a little lost. I feel there is a need to review the manuscript to identify where text could be reduced and still maintain meaning. For example this could be replaced by: "In terms of expression of separation-related behaviours during lockdown, significantly more respondents who owned dogs with pre-existing separation issues reported an increase compared to those that did not." 

 Table 3: I note the test statistic has ***, this should be explained (ie what do these asterisks mean?) I assume they show the P-value?

L282: what is meant by "did not cause non-convergence"?

L292: significance is often"less than or equal to" so here (and elsewhere) 0.05 could be reported as significant.

L388-398: these sentences about links between social cues and self-injurious behaviour don't seem to follow, the link is a little unclear. Could it be argued that SIB appears linked because it is the most extreme expression and therefore would follow a flow in development (ie dogs that reach the level of SIB will already have passed through tiers of lower-level signalling)?

Author Response

.

L21/36: the term "virtual absence" is odd, even with the e.g. I find it a little difficult to understand, possibly due to my own unfamiliarity. Is it "intended absence"?

We have corrected this to “owner attempts to go or is out of sight”, thank you.

L23: dog's

Fixed, thank you.

L65: maybe (or just may)

Fixed, thank you.

L73: worsen rather than deteriorate

Fixed, thank you.

L83-93: This is a rather lengthy report of a single paper, could it be better integrated within the intro? It would also be good to see a general intro to separation issues and their impact on owners and dogs. It needn't be exhaustive.

Thank you for the suggestion. We considered introducing more information about separation related issues, but do not want to conflate the syndrome (which focuses on the occurrence of signs associated with the absence of an owner) with specific diagnoses featuring these signs. Accordingly we simply want to focus on what owners see in relation to signs and not speculate about their clinical interpretation at this time. We have focused on the particular paper by Harvey et al., as it is the only other longitudinal study and thus of both topical and methodological significance to the current work.  We therefore believe it is important to give it specific emphasis.

L85-87: a little confusing as written. "However, of dogs with pre-existing separation-related behaviour problems, approximately 50% ceased to display them during lockdown" (I assume because the triggers were absent/reduced?).

Thank you, we have corrected this to “by contrast nearly half of dogs showing signs of these problems before lockdown ceased to display them during lockdown”.

L97: dataset

Fixed, thank you.

L97-99: this is a long sentence, I would suggest breaking it up for clarity. ("over time. This was based" "during the pandemic. Especially so given"). I would delete "because of the pandemic",  it is tautological.

Thank you, we have corrected this to “Our second aim was to use our unique longitudinal dataset to compare the emergence of potentially problematic behaviour in dogs over time. This comparison was based on both whether owners were working from home and the time the dog was left alone during the pandemic, given the increase in working from home at this time and associated disruption to normal routines.”

In M&Ms questionnaire part 2 is it possible to make reference to the behaviours outlined in Table 2 rather than listing them? Otherwise, the text is quite dense. (e.g. L106 The occurrence of 8 behaviours (see table n) were rated... etc.)
L125: Similarly, could the 16 behaviours be referred to a table (in this case 4)? Retaining those that were removed from analysis through non-occurrence in text.

Thank you, we have changed this to include a new table where the variables are more clearly laid out and to remove some of the denser text.

L148-188: The analyses are rather hard to follow, is it possible to break them down, possibly by simply bullet-pointing each step. Long paragraphs of multiple steps can be quite confusing.

Thank you! We have attempted to break this up to make it easier to follow.

I note that the sample was global. How was this addressed based on international differences in lockdown across the 8 months? Was location included in the analyses as a variable? This is perhaps a substantial limitation and the data on respondents should provide this info, assuming respondent data were collected.

We agree! While location was not included in the final analysis, as we had asked if they had exited lockdown and calculated the week in lockdown from their original date of staying home in Part 1, we determined the duration of their time in lockdown from those dates. We have added this to the manuscript as “As the survey was international and thus included a range of different starts for the lockdown, the period the dogs had been in lockdown was calculated for each of the participants using their answer to the date that they started to work from home, or if not working from home, the date the lockdown began in their region to the date the survey was taken.”

L 223-226: The writing in these sections is rather convoluted and so the meaning is a little lost. I feel there is a need to review the manuscript to identify where text could be reduced and still maintain meaning. For example this could be replaced by: "In terms of expression of separation-related behaviours during lockdown, significantly more respondents who owned dogs with pre-existing separation issues reported an increase compared to those that did not."

Thank you! We have revised the language here to “In terms of expression of separation-related behaviours during lockdown, significantly more respondents who owned dogs with pre-existing separation issues reported a decrease compared to those that did not. In particular, dogs with pre-existing issues of destruction of windows or doors, destruction of personal items, house-soiling, chewing if crated or self-injurous behaviour saw greater decreases in separation related problems, while those that did not show excessive drooling also had greater decreases in SRB problems.  Thus, dogs that excessively drooled pre-COVID did not show the same decrease in problematic behaviour when left alone as dogs that had other signs of destructiveness pre-COVID.”. 

 Table 3: I note the test statistic has ***, this should be explained (ie what do these asterisks mean?) I assume they show the P-value?

We have now replaced these with the p-values. Thank you for the suggestion. *** had been used to mean “p<0.001” but we realised that including the p-values as numerals is more valuable here.

L282: what is meant by "did not cause non-convergence"?

In a number of our models, a variable would be a perfect predictor. As this was not one, we have added the explanation that this variable “thus was not a perfect predictor, while explaining significant variation in the model”.

L292: significance is often "less than or equal to" so here (and elsewhere) 0.05 could be reported as significant.

Thank you for the suggestion. We have altered this to “was significant at p = 0.050” as there is still some debate about the accepted significance of 0.050 and we are more comfortable with the more conservative language here.

L388-398: these sentences about links between social cues and self-injurious behaviour don't seem to follow, the link is a little unclear. Could it be argued that SIB appears linked because it is the most extreme expression and therefore would follow a flow in development (ie dogs that reach the level of SIB will already have passed through tiers of lower-level signalling)?

Thank you we have revised the wording. We agree that the dogs pass through stages of yawning and blinking before progressing to self-injury. “Interestingly, self-injurious behaviour pre-COVID was  frequently a significant predictor was predictive of several related behavioural issues during lockdown: shadowing, destroying personal items, blocking attempts at separation and multiple stress related behaviours, including stretching and yawning. Thus, yawning and stretching may be indicative of social stress [27], although they are different to the signs reported to occur in inter-dog interactions [28]. This potentially indicates a relationship between social stress and self-mutilation, potentially as a form of redirected aggression [29] as has been reported in humans [30,31]. The association with yawning is particularly intriguing given its potential relationships with social cognition, e.g., the shared contagion suggesting empathy [32]. “

Reviewer 2 Report

General and specific comments

Taking the opportunities of drastic changes of ways of life due to COVID pandemia is the main positive aspect of this paper. It may appears paradoxical that separation related problems in dog-owner relationship is not solved when owners stay at home. However the good point of taking opportunities of a large scale experiment is totally hindered by the style of the paper. It is more than difficult to read and the potentially interesting and/or puzzling results are deeply burried into indigestible wording. Sentences are unreadable and quite often much too long (e.g L 94-96)..

In addition the tables are poorly or not referenced in the text and lack a real title explaining its rationale. In the downloaded document, it is quite difficult to differentiate table captions from the text.

Whereas many tables are like the text very difficult to understand, some results might be presented in a figure, such as results in paragraph 3.1.2, instead of clumsy wording.

In addition, readers lack how variables are collected. It seems that they are not mutually exclusive though treated as such. This is the case especially for vocalizations in table 3. Asterisks are not «  P-values ». P-values are numbers ! It lacks also the kind of test that is used and importantly the degrees of freedom. In table 2 why the sum of percentages for «Excessive drooling/salivating » equals 22,1 instead of 100.

To use sophisticated statistics to analyze responses (difficult to called them « data ») to an online questionnaire quite long and detailed, is always puzzling. How are data about dog behaviors collected when the dog is alone at home ? How are changes in dog behaviors reliable ? The impressive size of the sample do not confer reliability to behavioral variables.

This paper needs a huge amount of clarification, simplification before being reconsider for publication.

Author Response

The authors would like to thank the reviewers for their constructive and helpful comments. We have now incorporated these into the manuscript and feel it is much improved.

Taking the opportunities of drastic changes of ways of life due to COVID pandemia is the main positive aspect of this paper. It may appears paradoxical that separation related problems in dog-owner relationship is not solved when owners stay at home. However the good point of taking opportunities of a large scale experiment is totally hindered by the style of the paper. It is more than difficult to read and the potentially interesting and/or puzzling results are deeply burried into indigestible wording. Sentences are unreadable and quite often much too long (e.g L 94-96)..

We have now revised many of the sentences in the text and hope this version is more readable.

In addition the tables are poorly or not referenced in the text and lack a real title explaining its rationale. In the downloaded document, it is quite difficult to differentiate table captions from the text.

Apologies – we have been more careful in our presentation in the revised document.

Whereas many tables are like the text very difficult to understand, some results might be presented in a figure, such as results in paragraph 3.1.2, instead of clumsy wording.

While we recommend the value of this aesthetic, data sharing is important in science. In the interests of transparency and data specificity, we feel it is best to include the data in the paper. We have also included some of the data as figures in the supplementary and the raw data can always be used in this way, but the reverse is not possible to do accurately. Accordingly we have kept the data in tables.

In addition, readers lack how variables are collected. It seems that they are not mutually exclusive though treated as such. This is the case especially for vocalizations in table 3.

We have revised the text to make it clearer how questions were originally posed and hope this makes it easier to follow. The vocalisations were collected as individual behaviours, e.g., whimpering separate to howling, but then grouped into more general composite behaviour categories.

Asterisks are not «  P-values ». P-values are numbers ! It lacks also the kind of test that is used and importantly the degrees of freedom.

Thank you for the suggestion, we have now corrected the asterisks to p-values as numbers, instead of presenting them as p <0.001 == ***. We have also added the type of test used, Mann-Whitney, which does not typically present d.f. in results.

In table 2 why the sum of percentages for «Excessive drooling/salivating » equals 22,1 instead of 100.

Thank you for the catch! This was a transcription error on our part as the value for “Never” should have been 87.7% not 9.9%, which is the correctly given value in the next column.

To use sophisticated statistics to analyze responses (difficult to called them « data ») to an online questionnaire quite long and detailed, is always puzzling. How are data about dog behaviors collected when the dog is alone at home ? How are changes in dog behaviors reliable ? The impressive size of the sample do not confer reliability to behavioral variables.

We understand the reviewer’s point here but surveys are reliably used in this context and there has been a study of owner reliability (Ley et al. 2009 - https://doi.org/10.1016/j.applanim.2009.02.027).  It has been found that large datasets reduce random error from individual level unreliability, and that trends are unlikely to be affected in large datasets. Of course, there is also a chance of systematic error, but we feel that while surveys are not absolute, they are a reasonable approach and one that is widely accepted.

This paper needs a huge amount of clarification, simplification before being reconsider for publication.

We hope our revised version is clearer and answers the reviewer’s concerns.

Reviewer 3 Report

In the present work the authors carried out a survey a longitudinal survey in dog owners, who radically changed their habits During the COVID-19 pandemic, in order to assess the potential impact that such restrictions in daily routine had upon dog behavior. They documented that the dogs, who experienced separation-related issues even before the pandemic lockdown, had more negative symptoms, when compared to control animals. The paper sounds quite interesting, considering the huge number of behavioral parameters addressed both in dogs and their owners, although the more severe symptoms in those dogs might be obvious.

However, some minor considerations are listed below:

1) It is not so clear the real impact of separation-related issues upon the main features analyzed. The authors are strongly advised to report the main findings into discussion.

2) What about the beneficial effects (it is not so clear if any) of the pre-COVID separation-related problems experienced during the pandemic lockdown?

3) What did the authors mean with “out to work”? Please state better in the table description, to make it more understandable.

4) since there are several spelling/grammar mistakes, a thoughtful English editing is suggested.

Author Response

We would like to thank the reviewer for their very helpful suggestions and hope that we have addressed them all effectively below.

In the present work the authors carried out a survey a longitudinal survey in dog owners, who radically changed their habits During the COVID-19 pandemic, in order to assess the potential impact that such restrictions in daily routine had upon dog behavior. They documented that the dogs, who experienced separation-related issues even before the pandemic lockdown, had more negative symptoms, when compared to control animals. The paper sounds quite interesting, considering the huge number of behavioral parameters addressed both in dogs and their owners, although the more severe symptoms in those dogs might be obvious.

However, some minor considerations are listed below:

  • It is not so clear the real impact of separation-related issues upon the main features analyzed. The authors are strongly advised to report the main findings into discussion.

We have attempted to include a clearer statement of the main findings in the discussion. The first paragraph now reads thus:

“Pre-COVID, approximately 20% of dogs in our sample showed signs of separation-related vocalisation at least quite frequently when left alone for more than an hour and 15% of dogs showed running at windows, but only about 5% of dogs showed some form of destructiveness or elimination in these circumstances. COVID lockdowns generally increased the prevalence of dogs presenting separation-related problem behaviours (Table 5) and this increase was generally more often significant in dogs with pre-existing problem behaviours, though some dogs did show a significant decrease in destructive behaviours compared to baseline.”

We have also revised the discussion to better explore our results and contextualise.

2) What about the beneficial effects (it is not so clear if any) of the pre-COVID separation-related problems experienced during the pandemic lockdown?

We found few beneficial effects except on reduction of destruction of personal items, so we have included this more explicitly in the discussion.

3) What did the authors mean with “out to work”? Please state better in the table description, to make it more understandable.

Thank you, we have changed this to “working outside of the home”.

4) since there are several spelling/grammar mistakes, a thoughtful English editing is suggested.

Thank you, we have checked the spelling and revised the language throughout.